# The Evaluation of IDEAL-REACH Program to Improve Nutrition among Asian American Community Members in the Philadelphia Metropolitan Area

**DOI:** 10.3390/ijerph16173054

**Published:** 2019-08-23

**Authors:** Grace X. Ma, Lin Zhu, Steven E. Shive, Guo Zhang, Yvette R. Senter, Pablo Topete, Brenda Seals, Shumenghui Zhai, MinQi Wang, Yin Tan

**Affiliations:** 1Center for Asian Health, Lewis Katz School of Medicine, Temple University, Philadelphia, PA 19140, USA; 2Department of Clinical Sciences, Lewis Katz School of Medicine, Temple University, Philadelphia, PA 19140, USA; 3Department of Health Studies, East Stroudsburg University, East Stroudsburg, PA 18301, USA; 4Division of Nutrition, Physical Activity and Obesity, NCCDPHP, CDC, Atlanta, GA 30333, USA; 5ICF International, Atlanta, GA 30329, USA; 6Department of Behavioral and Community Health, School of Public Health, University of Maryland, College Park, MD 20742, USA

**Keywords:** community intervention, Asian American, nutrition, health disease prevention

## Abstract

*Objective* Asian Americans’ food purchasing, cooking, and eating patterns are not well understood. Greater insight into these behaviors is urgently needed to guide public health interventions of dietary behaviors in this population. The present study aims to examine the effects of a community-level intervention on food purchasing and preparation, nutrition knowledge, and health awareness in Asian Americans. *Methods* From 2015 to 2017, we conducted the Improving Diets with an Ecological Approach for Lifestyle (IDEAL-REACH) intervention to increase access to healthy food or beverage options for the Asian-American population in the Philadelphia metropolitan area. Participants (1110 at pre- and 1098 at post-assessment) were recruited from 31 community-based organizations (CBOs). We assessed Asian Americans’ dietary behaviors, nutrition knowledge, and awareness of heart health. *Results* The results of pre-post intervention comparisons showed that the IDEAL-REACH intervention was successful in promoting whole grains consumption, reducing sodium consumption, and raising knowledge and awareness related to nutrition and heart health. *Conclusions* To our knowledge, this is one of the first initiatives in the U.S. to engage CBOs to promote healthier dietary behaviors. The findings show that CBOs serve as a powerful platform for community-level interventions to improve healthy nutrition behaviors in Asian-American communities.

## 1. Introduction

Obesity rates are on the rise among Asian-American adults and children, and as a consequence these populations are also experiencing an increase in incidences of obesity-associated conditions including diabetes, hypertension, and cardiovascular diseases [1,2,3]. Dietary change is a major contributing factor behind these trends. Traditional Asian cooking is marked by a high intake of vegetables, fruit, and legumes and by a low intake of fat and meat. Low in fat and high in fiber, traditional Asian diets are associated with a reduced risk of heart disease [3,4]. However, current Asian-American diets in the US are characterized by high sodium content, high carbohydrate content, and heavy use of cooking oils [5,6]. With the global popularization of processed food and fast food, the diets of populations in Asia and in the US are changing. Specifically in the US, Asian Americans are eating more processed food and fast food and experience poorer nutrient intake relative to Whites [7,8]. A study of Korean-American adolescents showed that half of the adolescents’ energy came from carbohydrates, 53.2% for boys and 54.1% for girls [9]. Evidence also suggests that Asian-American children are less likely to eat vegetables and fruits than White and Latino children [10]. While the traditional Asian diet is associated with positive health effects, the current Asian-American diet seems to have unfavorable effects on health outcomes, owing in particular to high sodium and carbohydrate intake and a lack of fiber.

### 1.1. Dietary Factors and Cardiovascular Disease

For Asian Americans, heart disease is the second leading cause of death [11]. Studies have identified dietary factors as significant predictors of cardiovascular disease (CVD) risk factors among Asian Americans [12,13]. There is a link between high salt intake and cardiovascular diseases, such as high blood pressure [14]. People living on the Northern Islands of Japan eat the most salty food and have the highest incidence of hypertension, while Japanese individuals who have low salt intake show virtually no traces of hypertension [15]. Researchers estimate that the average Asian American sodium intake is 4600 mg daily, obtained from table salt, soy sauce, and canned and processed foods [16]. Asian diets also include unhealthy cooking oils that contain trans and animal fats, which are associated with weight gain, increased risk of cardiovascular disease, and insulin resistance contributing to type 2 diabetes [17]. By contrast, low-fat vegetarian diets have been found to lower blood pressure and increase longevity [18].

Diet affects cardiovascular disease risk. A growing body of literature shows that increased intake of whole grain foods, reduced intake of sodium, and use of certain plant-based cooking oils, such as olive or canola oil, can have significant health benefits in the general population. Refined grain and whole grain foods have vastly different impacts on health. For example, a meta-analysis of prospective studies indicated that a higher white rice intake is associated with an increased risk of type 2 diabetes, especially among Asian populations [19]. Meanwhile, adding two servings a day of whole grain intake is associated with a 21% reduction in risk of diabetes [20]. The consumption of whole grain foods over a 12-week period was found to reduce blood levels of insulin and triglycerides in individuals with metabolic syndrome compared to a control group that consumed processed cereals [21]. Further, in a review of 15 studies which included data on 119,829 people ages 13 and older, researchers found that body mass index (BMI) was lower when people consumed more whole grains compared to those who consumed fewer whole grains. Higher intake of whole grains was also associated with increased dietary fiber and reduced saturated fat intake [22].

Adherence to the Dietary Approaches to Stop Hypertension (DASH) eating plan, which encourages a diet high in fruits, vegetables, low-fat dairy products, whole grains, poultry, fish, and nuts, can lower systolic blood pressure by 5 to 6 mm Hg [23]. When the DASH diet is used in combination with sodium reduction, even greater reductions can be achieved. For example, when sodium intake was lowered to 1500 mg, below current recommendations, a reduction in blood pressure of 8.9/4.5 mm Hg was obtained (7.1/3.7 mm Hg in non-hypertensive participants, and 11.5/5.7 mm Hg in hypertensive participants) [23]. This DASH diet plus sodium reduction was assessed in a clinical environment, where participants consumed prepared meals. Research showed that a reduction in salt by 3 g per day would lead to significant decreases in coronary heart disease, strokes, myocardial infarctions, and deaths from any cause yearly [24]. Hence, reduction in salt can lead to a decrease in blood pressure of sufficient magnitude to substantially reduce the risk of adverse cardiovascular events [25].

### 1.2. Multi-Level, Community Led Interventions

Interventions actively involving the target community and employing multi-level salt reduction intervention strategies have documented effectiveness in reducing dietary salt intake, urinary sodium excretion, blood pressure, and the amount of sodium in purchased foods and improving health outcomes [26,27,28,29,30,31,32]. In this project, community-based organizations (CBOs), such as churches, temples, and senior care centers, are essential settings for environmental and organizational change, especially for underserved populations. CBO representatives played key roles in all phases of the planning, implementation, and evaluation of this project, consistent with CBPA principles. During the study conceptualization, design, implementation, and evaluation stages, CBO leaders, chefs, and other community stakeholders were actively engaged. Specifically, the research team worked closely with CBO leaders, chefs, staff, and other stakeholders to develop, test, and revise nutrition guideline messages, easy-to-practice tips, social media articles, and evaluation survey questionnaires. In addition, CBO leaders, chefs, and staff played active roles in designing and organizing the food tasting events and technical assistance on food preparation and cooking.

In previous studies, health promotion by CBOs has been associated with decreases in salt intake (15%), increases in the intake of fiber (42%) and other healthy ingredients (43%), and improvement in other health behaviors [33,34,35]. Senior care centers provide daily meals to adult Asian Americans, assisting Asian seniors who may face social isolation and poor nutritional status [36]. These CBOs provide an opportunity to improve nutritional intake. The findings in previous studies indicate that if participants can increase their intake of plant-based foods and reduce salt and fat intake, then their risk of hypertension, stroke, diabetes, and obesity will decrease and they will benefit from positive effects on the cardiovascular system in general [36].

Positive changes in knowledge, attitudes, behaviors, physiological measures, and cost-effectiveness have also been reported for salt reduction multi-level, community initiatives [37]. Community-level interventions, which enhance public awareness about the importance of reducing sodium in foods may lead to greater demand for lower sodium products and motivation for lowering salt in cooked meals and increased use of lower salt products by families and community organizations. To reduce sodium chloride in foods, one effective approach is to substitute sodium with other items, such as spices, low- or no-sodium sauces (e.g., fish sauce), or other types of salts containing calcium or potassium [38]. These substitutes can be made with little or no significant loss of taste. The main barriers to maintaining a low-sodium diet are taste, lack of acceptability of reduced sodium foods, lack of control over food preparation in out-of-home meals, and difficulty identifying suitable snacks. Facilitators to obtaining reduced-sodium meals include meal planning and preparation from scratch, using flavor replacements, reading food labels, and receiving social support and motivation at weekly clinics, especially if a person has existing heart failure and hypertension [37]. Community-level interventions to reduce sodium target individuals to add less salt “at the table” and during food preparation through health education and focus on national policy changes through setting targets or legislation for sodium content in processed foods [36]. These interventions are crucial, given the sodium overconsumption by Americans.

The purpose of this study is to examine the effects of the IDEAL-REACH (Improving Diets with an Ecological Approach for Lifestyle) intervention by measuring CBO members’ food purchasing and preparation practices, nutrition knowledge, and awareness of health promotion campaigns. The CBOs enrolled in this study serve large, Asian populations and senior citizens, who are often dependent on such organizations for many of their meals. The intervention centered on improving meal plan preparation and increasing knowledge to reduce the use of sodium and unhealthy oils and increase consumption of whole grain products.

## 2. Materials and Methods

### 2.1. IDEAL-REACH Program

The purpose of the IDEAL-REACH project is to increase access to environment with healthy food or beverage options for the underserved Chinese, Vietnamese, Korean, and Filipino American communities in the greater Philadelphia metropolitan area. The ultimate goal is to reduce the risk factors for diabetes, heart disease, hypertension, and stroke and improve overall health of Asian Americans. The IDEAL project, supported by CDC REACH (Racial and Ethnic Approaches to Community Health) funding, provides a multi-faceted intervention including health education for CBOs, media stories on healthy nutrition and success stories, and product placement and expansion for Asian Groceries tailored to each community utilizing the Community-Based Participatory Approach (CBPR) and the Social Ecological model [39]. Our goal was to reach individual community members, community organizations serving the community, and grocery stores. Because Asian ethnic communities are dispersed across the Philadelphia metropolitan area, some community members traveled long distances to attend events and CBO activities as well as to shop at Asian stores. Hence, community, for this part of the study, is based on membership in a CBO and language and ethnicity self-identification. The IDEAL CBO intervention components included (1) distributing nutrition education brochures and displaying posters in both English and Asian ethnic languages; (2) providing healthy food preparation and cooking technical assistance to CBO leaders, chefs, staff, and community members; (3) hosting healthy food tasting events to introduce low-sodium recipes and other healthy food choices; and (4) providing technical assistance on nutrition guidelines to CBO leaders, chefs, staff, and community members. The community-level media communication intervention engaged multiple media campaign activities through several influential local Asian newspapers and media outlets, CBO newsletters, and popular social media tools. Cultural and linguistic appropriate nutrition guideline messages and easy-to-practice tips were disseminated using these communication channels. Specifically, we distributed nutrition education brochures and displayed posters in all participating CBOs, provided technical assistance on food preparation, cooking, and nutrition guidelines in all CBOs, and held healthy food tasting events.

Stratified-cluster proportional sampling technique was used to recruit CBOs, while convenient sampling was used to recruit participants in each CBO [40]. A current listing of Asian-American CBOs (*n* = 88), such as churches, temples, and senior daycare centers, in the seven counties of the Delaware Valley region of Pennsylvania and southern New Jersey were identified by the Center for Asian Health staff and the Asian Community Health Coalition. These 31 organizations were located in geographic areas that maximized the coverage of Asian American communities (Chinese [6], Korean [12], Vietnamese [9] and Filipino [4]), ages, and socioeconomic status. A total of 66 organizations participated over the three years, including 16 Chinese, 29 Korean, 17 Vietnamese, and 4 Filipino organizations. The selected organization clusters were stratified based on the four race/ethnicity or language groups. A proportional allocation procedure of assigning the sample size proportionally to the subgroups’ size was used [41]. The overall CBO member sample consisted of 1110 for the baseline survey and 1098 for the 12-month follow-up survey.

### 2.2. Measurements

A 46-item questionnaire baseline survey and a 48-item 12-month follow-up survey questionnaire were developed. At the time, there were no studies tailored for Asian American diets focusing on salt, oil, and whole grains, hence the questionnaire was developed either based on available questions from other national surveys or new questions reviewed and approved by an advisory group for each community. A pilot test was conducted to validate the instrument and to verify data collection methods. The appropriateness of the questionnaire format, content and face validity, and length of time to take the survey were determined in a pilot test with 15 Asian-American adults, representing the various language groups, and were revised based on participant feedback. The final questionnaire included: demographics, food purchases, food preparation/cooking practices, nutrition knowledge, program promotion, and use of media and social media. Average time for completion of the questionnaire was 20 to 25 min.

Outcome Measures. At the 12-month follow-up, three outcome measures were assessed: the purchase of low-sodium foods, the purchase of healthier oils, and the purchase of whole grain products in the past 6 months. Answers were dichotomous, with 0 being “no” and 1 being “yes”.

Food Purchases and Preparation Practices. Because one important component of our intervention was promoting low-sodium soy sauce to replace regular soy sauce, we asked at baseline and 12-month follow-up about the size of the bottle of regular soy sauce purchased and how often they purchased soy sauce (in months). We also assessed at both time-points the bag size of white and brown rice purchased, as well as the purchasing intervals. Furthermore, we asked respondents about the specific types of oil they purchased, including canola, sesame, corn, olive, and vegetable oil. We asked about the container size and intervals of their oil purchase. Respondents were also asked if they checked nutrition labels for specific contents, including sugar, sodium, calories, fiber, unsaturated fat, trans fat, and saturated fat content of the food when making purchases. We asked about frequency of shopping at Asian supermarkets, and the number of Asian food markets participants shop in. Regarding food preparation behaviors, we measured the number of people respondents usually cooked for, whether they measured salt and oil when cooking, and whether they tried to use less sodium, less oil, and more whole grains in their cooking. Last but not least, respondents were asked whether they add salt to their food at the table. The purchasing intervals were winsorized at the 95th percentile to reduce the effect of possibly spurious outliers (e.g., one respondent reported purchasing 1 bottle of soy sauce every 24 months).

Knowledge of Nutrition and Health Effects Related to Sodium, Oil, and Fiber. We used eight questions to assess CBO member participants’ knowledge related to physical activity and health effects of salt, oil, and whole grain intake. Specific questions included recommended frequency of physical activity, the health benefits of exercise, the recommended amount of salt intake, the main source of salt in an average American diet, and the health benefits of a high fiber diet. For the questions with only one right answer, e.g., “one teaspoon” being the correct answer to “the recommended level of salt in diet for a healthy adult each day”, participants’ answers were dichotomized as 1 “right” or 0 “wrong.” For the questions with multiple right answers, e.g., “more energy”, “decrease colon cancer risk”, “lower spikes in blood sugar”, and “easier to go to the toilet” being the right answers of “the health benefits of a high fiber diet,” a score was computed for the question and was then linear-transformed to a scale of 0 to 1. The answers to all eight questions were then summed up to make up the total knowledge score. The score ranged from 0 to 7, with a higher numeric value indicating a higher level of nutrition-related knowledge.

Exposure to the IDEAL Campaign. Participants were also asked if they use Asian language newspapers and whether they saw any signs or messages that promoted reduced sodium intake, the use of better oil, and increased intake of whole fiber foods.

Demographics. These items included zip code, race/ethnicity, age, gender, years lived in the U.S., education, employment status, English fluency, language spoken at home, current insurance status, having a regular physician, and perception of their health status compared to others.

In the follow-up survey, participants were asked if they purchased any products within the last 6 months that contained reduced sodium, healthier oils, and more whole grains. In addition, if they responded positively to the purchase of the items, they were asked if their family liked them, if they used the product in cooking, and would they use it again in the future.

### 2.3. Data Collection Procedures

Standard questionnaire training was provided to all questionnaire administrators as well as to onsite bilingual translators. The questionnaire was distributed to CBO leaders, who then administered them to their members prior to and after the intervention. Center for Asian Health (CAH) staff provided technical assistance to the bilingual staff of the CBOs when needed and were often available to assist with questionnaire distribution and collection. Participants had the option of responding to the questionnaire in English or in Chinese, Korean, and Vietnamese. Participants completed the questionnaires, which were collected by the CBO leaders and given to CAH staff.

### 2.4. Data Analysis

Data were analyzed with Stata 14 [42]. The study used descriptive statistics to analyze food purchases, food preparation/cooking, nutrition knowledge, program promotion, and use of media and social media. Descriptive statistics were calculated and chi-square tests and Student’s t-tests was used to examine the changes from baseline to 12-month follow-up for repeated variables. To examine the association between three healthier purchasing behaviors and exposure to the IDEAL campaign, nutrition knowledge, and sociodemographic factors, three separate binary logistic regression models were built. The regression coefficients (log odds) and goodness-of-fit statistics were presented for each model.

## 3. Results

Descriptive statistics of sociodemographic, immigration-related, and health-related factors at baseline and 12-month follow-up are presented in Table 1, along with chi-square test results of the comparison between baseline and 12-month follow-up. Overall, there were great similarities between the baseline and the 12-month follow-up questionnaires samples on race/ethnicity and gender composition, educational attainment, employment status, and the proportion of having health insurance or a regular physician (*p* < 0.05). However, participants in the 12-month follow-up sample were younger, had slightly higher English proficiency, were more likely to speak both English and a native Asian language at home, and had better self-rated health than the baseline samples.

Table 2 shows the distribution of food purchases and preparation practices at baseline and 12-month follow-up. First, the average regular soy sauce purchasing interval at the 12-month follow-up survey was 2.02 months, which was significantly longer than the baseline interval of 1.86 months (*p* < 0.05). Despite the insignificant change in the size of bag of white rice purchased, a significant increase occurred in the purchasing intervals (1.94 to 2.11 months, *p* < 0.01). The percentage of participants who did not eat brown rice decreased notably (26.74% to 17.95%), and large bags of brown rice (40+ pounds) became less popular (3.53% to 3.17%), although the purchasing interval did not change significantly. Regarding the types of oil purchased, canola (27.66% to 36.56%) and sesame (13.96% and 19.10%) oil gained popularity, while the other three types did not show significant change (*p* > 0.05). Although the size of the oil container did not change significantly from baseline to 12-month follow-up, the purchase intervals increased (1.91 to 2.31) Furthermore, significantly higher proportions of participants reported that they read nutrition labels for calories (*p* < 0.05), fiber (*p* < 0.001), and trans fat (*p* < 0.05) content when making food purchases.

Second, regarding food preparation behaviors, we found no significant changes in whether or not participants measured salt when cooking, although there was a slight increase in the percentage of those who did not measure oil in cooking (76.25% to 80.27%, *p* < 0.05). At both baseline and 12-month follow-up, the great majority (>80%) of the participants said they were trying to use less sodium and oil, and more whole grains in their cooking; yet, the only significant reported change was for whole grain intake, with an increase from 66.35% to 73.13% (*p* < 0.001).

Third, we found a positive change in salt consumption behavior. Specifically, a lower proportion of participants said they “always” added salt to their food at the table (14.02% to 9.15%) and a higher proportion reported not adding salt (30.98% to 36.97%). The overall change of this behavior was statistically significant (*p* < 0.001).

The descriptive and comparative statistics on the eight knowledge items and the total knowledge scores are presented in Table 3. A statistically significant improvement was seen from baseline to the 12-month follow-up for four items of knowledge: the amount of physical activity recommended for those with diabetes (*p* < 0.01), recommended daily sodium intake (*p* < 0.001), the main source of sodium in a typical American diet (*p* < 0.05), and the healthiest type of fat (*p* < 0.001). The overall knowledge score, presented at the bottom of the table, is significantly (*p* < 0.001) higher at the 12-month follow-up (4.68) than at baseline (4.32).

Descriptive and comparative statistics of exposure to the health promotion messages are presented in Table 4. The proportion of respondents who reported noticing signs that promoted healthier oil, whole grain, and low-sodium products increased significantly from baseline to 12-month follow-up (*p* < 0.001). The proportion of respondents who reported noticing signs that promoted healthier food in general did not increase significantly (*p* > 0.05).

The descriptive statistics of the three outcome measures, along with related attitude and intention are shown in Table 5. At the 12-month follow-up, 40.4% of respondents reported having purchased low-sodium products, 37.5% purchased whole grain products, and 37.8% purchased healthier oil products in the past 6 months. Among those who reported having purchased low-sodium products in the past 6 months, the majority said that they used the products in their cooking (71.36%), that they and their family liked the products (77.09%), and that they would continue to use the products (77.09%). Similarly, among those who reported having purchased healthier oil in the past 6 months, the majority said that they used the healthier oil in their cooking (68.71%), that they and their family liked it (78.21%), and that they would continue to use it (82.51%). Among those who reported having purchased whole grain in the past 6 months, more than three-quarters said that they used the whole grain in their cooking (78.59%), that they and their family liked it (82.02%), and that they would continue to use it (83.38%).

Three separate multivariate logistic regression models were constructed to examine the association between three outcomes: purchasing reduced sodium products, purchasing whole grain products, and purchasing healthier oil. Several predictors were considered, including exposure to IDEAL health promotion signs, nutrition knowledge, and sociodemographic variables. The results are presented in Table 6. In the first model, a higher knowledge score (β = 0.24, *p* < 0.001), having noticed signs that promoted healthier foods (β = 0.59, *p* < 0.01), Filipino ethnicity (vs. Chinese, β = 2.14, *p* < 0.001), other Asian ethnicity (vs. Chinese, β = 1.74, *p* < 0.05), and female gender (vs. male, β = 0.49, *p* < 0.01) were significantly associated with a higher likelihood of purchasing low-sodium products in the past 6 months. In the second model, a higher knowledge score (β = 0.33, *p* < 0.001), Filipino ethnicity (vs. Chinese, β = 2.15, *p* < 0.001), and female gender (vs. male, β = 0.57, *p* < 0.01) were significantly associated with a higher likelihood of purchasing whole grain products in the past 6 months. Korean respondents were less likely than the reference group (Chinese respondents) to have purchased whole grain products. In the third model, a higher knowledge score (β = 0.35, *p* < 0.001), noticing signs that promoted healthier oil (β = 1.58, *p* < 0.001), Filipino ethnicity (vs. Chinese, β = 1.93, *p* < 0.001), and female gender (vs. male, β = 0.62, *p* < 0.01) were significantly associated with a higher likelihood of purchasing healthier oil products in the past 6 months. Altogether the predictors accounted for 13%, 19% and 18% of the variance in the outcome variables in the three models, as indicated by the R-square statistics.

## 4. Discussion

This study evaluated the intervention effects of the IDEAL-REACH project for Asian American members of Community Based Organizations. Purchasing, cooking, and consumption behaviors in Asian-American communities were investigated between baseline and a 12-month follow-up. Several predictors of healthier behaviors were identified. Our findings shed light on patterns of related behaviors in Asian-American communities and generate implications for future public health campaigns, policies, and research.

### 4.1. Food Purchases and Preparation Practices

Across the board, members of Asian CBOs reported that they wanted to improve their diets with better cooking, preparation, and eating practices for salt, oils, and whole grants. Community members reported longer purchasing intervals at the 12-month follow-up compared to baseline for soy sauce, white rice, and oil (Table 2). These findings suggest, given that the reported purchase size did not change, a decrease may have occurred, a sign of intervention effectiveness. Because regular soy sauce is a popular condiment that is high in salt even in the “lower salt” versions, even a slight decrease would contribute to overall sodium reduction in Asian communities. Similarly, for white rice and oil, these changes indicate healthier eating patterns.

We also found significant intervention effects in increasing the purchases of healthier food such as brown rice and canola and sesame oil, and improvements in behaviors like less added salt at the table which suggests increasing attention to calories, trans fats, salt, and fiber also evidenced by improvements in reported reading food labels. Respondents also noted good media coverage for healthy foods and noticing signs in stores. While healthy food is a common topic in the media, the signage in the Asian stores and stories in the local news supported by the intervention featured local success stories and were distributed in four different languages. While we cannot directly link these media interventions to these documented improvements in health behavior, this media effort plus the “buzz” from health education at the CBOs and in the stores with tasting events, likely improved healthy food awareness at a community level.

Multivariate logistic regression results identified several significant predictors of purchasing low-sodium, whole grain, and healthier oil products. Again, noticing signs that promoted healthier foods was significantly related to purchasing low-sodium and healthier oil products. Equally important, improvements in knowledge of nutrition and health effects of exercise, proves to be a significant predictor of all three outcome measures. Because the Asian diet is generally thought to be healthy, most Asian communities do not think about aspects of their diet that are unhealthy and do not consider changes that may have occurred in their diets and the diets of their children since moving to the US [43,44,45]. Changing these perceptions to improve knowledge of healthier eating and exercise must be a key component of effective health campaigns.

Despite the aforementioned positive changes in several behaviors, our study sample presented suboptimal sodium and fiber cooking behaviors, which concur with previous findings of dietary behaviors among Asian Americans’ high intake of sodium and refined grains [5,6,40]. More than half of the participants did not purchase any kinds of whole grains, and more than three-quarters did not measure salt or oil in their cooking, suggesting that their consumption of whole grain, salt, and oil were likely suboptimal. In addition, respondents’ health knowledge was less than ideal. Specifically, only about three-quarters of the respondents at the 12-month follow-up could correctly identify processed food as the main source of sodium in a typical American diet. Less than half of respondents could correctly identify monounsaturated and polyunsaturated fats as the healthiest type of fats. Knowledge on the health benefits of a high fiber diet was also lacking. Therefore, a need remains in Asian-American communities to enhance knowledge about nutrition and about how nutrition and health are related to sodium, fiber, and oil intake. Maybe even more important, Asian American communities need to be more aware of health risks related to diabetes, obesity, hypertension, and cardiovascular disease that can be prevented and controlled through healthy diet and exercise. Because Asian Americans develop such diseases at lower BMI levels and these diseases have traditionally been rare in the community, many community members do not take these health risks seriously [46,47,48,49,50]. Health education must be holistically addressing daily behaviors (e.g., not adding too much salt to food), success stories, general healthy nutrition, and the rise in disease in Asian populations.

### 4.2. Ethnic and Gender Considerations

Variations of purchasing behaviors manifested within the Asian-American populations. Specifically, Filipino community members purchased healthier food products, while other variables were held constant. Unseen from previous studies, this unique findings of dietary patterns and food purchasing behaviors among Filipino Americans was surprising. Filipino Americans have a notably higher prevalence of heart disease, diabetes, obesity, and hypertension than Chinese Americans, according to national and regional studies [51,52,53,54,55]. A lower level of physical activity is a major contributing factor to the high burden of cardiovascular disease among Filipino Americans [56]. Hopefully the healthier behaviors documented here will make a long-term difference in health outcomes.

Conversely, Vietnamese respondents were less likely to purchase whole grain products, which partially confirms previous findings of low dietary fiber intake among Vietnamese Americans [57]. Existing findings, though scanty, suggest a pattern of low-fiber, high-fat diet in Vietnamese Americans, especially among the younger generations [58,59,60].

Female respondents were more likely than male respondents to purchase low-sodium, whole grain, and healthier oil products, consistent with previous literature on the gender differences in dietary behaviors [61,62,63]. Men typically have a diet and purchasing behaviors that consists of fewer whole grains and vegetables and more fats and alcohol than women’s diets [61,64]. Women’s stronger belief in dietary guidelines and better knowledge in food and nutrition may be part of an explanation of their longer longevity [65]. However, gender differences in purchasing behaviors persist beyond the contribution of knowledge suggesting that other variables may be at play. Future studies should examine these gender differences and identify ways to improve healthy purchasing and dietary habits, especially for men. How social environment affects men’s health behaviors and outcomes must be considered in public health research of men’s health.

### 4.3. Community and Policy Implications

Findings from this study show that a multi-level, population-based intervention and participatory research approach with CBO involvement is an effective means of promoting reduced sodium intake, increased fiber intake, and the use of healthier cooking ingredients. CBO representatives were at the decision table throughout this project and continue as active supporters concerned to improve the health of their communities sustaining changes beyond this intervention project. The overall cost of this project, especially if changes are sustained, is small in comparison to the burden of care represented by increases in health problems linked to diet in Asian Americans. Previous studies have reported similar results [33,34,35,66,67,68]. Researchers particularly highlighted the importance of building coalitions, tailoring culturally specific messages, targeting captive audiences in the delivery of the intervention, and emphasizing community rather than individual norms [67,69,70]. This study contributes to the growing body of literature on community-based lifestyle intervention by expanding the focus to specific behaviors related to sodium, oil, and fiber intake in previously understudied Asian-American communities.

Whether or not the documented changes indicated in this study are sufficient to meet World Health Organization (WHO) targets for 2025 are important for the next step in evaluating nutrition programs [71]. More efforts need to be made to improve access to low salt alternatives for Asian American populations. Lower salt soy sauce remains high in sodium. Asian American cooking sauces (like shrimp sauce) and pastes are high in salt. At baseline, our assessment of whole grain options in the Asian American grocery stores revealed no high fiber noodles or wraps for dumplings. Although healthier oils were available, such oils were often in small bottles or few options, especially low-cost options. Our efforts to contact food producers and advocate for lower salt options were frustrating. 

Our participatory approach was able to achieve positive changes in several aspects of healthier food purchasing behaviors. Tracking to see how sustainable such changes are or if it takes longer for dietary habits to change would be a contribution to the field as our community partners indicate positive advances. Additionally, we did not examine clinical measures, such as BMI, blood pressure, or plasma glucose levels; hence our inability to assess the effects of interventions on health outcomes. Assessment of these factors in health outcomes is needed in future community- and population-based studies [67]. Furthermore, given that there were multiple exposure components to the campaign, such as taste testing, educational signs, cooking training, and newsletters, it is difficult to identify which specific intervention component had the most impact on the overall results. The comprehensiveness of the outreach and education may have contributed the most. Despite these limitations, this is one of the first initiatives to engage Asian-Americans CBOs to influence food purchasing and preparation practices and nutrition knowledge among their members to improve dietary intake of sodium and fiber/whole grains and to promote the use of healthier types of oil. Our findings show that CBOs serve as a powerful platform for community-level interventions to change health behaviors and promote healthy eating in Asian-American communities and have implications for future interventions and research.

## 5. Conclusions

The westernization of dietary patterns among Asian American and specifically Vietnamese immigrants has concerning health implications for health care access and costs as well as quality of life and shortened lifespans. Recent trends show significant increases in dietary related diseases which, if interventions are not forthcoming, can only be exacerbated. Tracking these trends and setting a direction for future research are critically important for addressing these health disparities. Health campaigns meant to decrease diabetes and obesity and improve cardiovascular health through diet should highlight messages of healthier food choices in general and healthy oil. Because knowledge was a consistent predictor of all three outcomes, future intervention campaigns and policies should focus on designing and delivering materials to raise awareness and knowledge of the nutritional facts and health impacts of specific food items.

## Figures and Tables

**Table 1 ijerph-16-03054-t001:** Sociodemographic, Language, and Health-Related Factors.

Characteristics	Baseline *n* = 1110	12-Month Follow-Up *n* = 1098	χ^2^ (*df*)
CBO ethnicity (sites)			
Chinese	6 (19.35%)	6 (19.35%)	
Korean	12 (38.71%)	12 (38.71%)	
Vietnamese	9 (29.03%)	9 (29.03%)	
Filipino	4 (12.90%)	4 (12.90%)	
Race/ethnicity			2.91 (4)
Chinese	264 (23.85%)	248 (23.24%)	
Korean	344 (31.07%)	360 (33.74%)	
Vietnamese	292 (26.38%)	270 (25.30%)	
Filipino	196 (17.71%)	174 (16.31%)	
Other	11 (0.99%)	15 (1.41%)	
Age			15.47 (3) ***
18–29	45 (4.16%)	73 (6.85%)	
30–44	222 (20.54%)	255 (23.94%)	
45–60	428 (39.59%)	419 (39.94%)	
>60	386 (35.71%)	318 (29.86%)	
Gender			7.44 (1)
Male	298 (28.11%)	346 (33.62%)	
Female	762 (71.89%)	683 (66.38%)	
Education			2.72 (2)
<High school graduate	253 (23.67%)	286 (26.75%)	
High School	260 (24.32%)	252 (23.57%)	
University and above	556 (52.01%)	531 (49.67%)	
Employment status			4.90 (4)
Employed	515 (47.47%)	557 (51.91%)	
Unemployed	79 (7.28%)	75 (6.99%)	
Retired	267 (24.61%)	234 (21.81%)	
Homemaker	195 (17.97%)	176 (16.40%)	
Student	29 (2.67%)	31 (2.89%)	
English proficiency			8.72 (1) *
Not at all/not well	675 (62.68%)	606 (56.42%)	
Well/fluently	402 (37.32%)	468 (43.57%)	
Language spoken at home			12.07 (2) *
English	74 (6.85%)	69 (6.41%)	
Native Asian language	780 (72.22%)	713 (66.26%)	
English and native Asian language both	226 (20.93%)	294 (27.32%)	
Has health insurance	925 (84.94%)	936 (86.99%)	1.52 (1)
Has a regular physician to visit	914 (84.16%)	912 (84.92%)	0.20 (1)
Self-rated health			5.73 (1) *
Excellent/good	869 (79.95%)	902 (83.90%)	
Poor/very poor	218 (20.06%)	173 (16.09%)	

* *p* < 0.05, *** *p* < 0.001.

**Table 2 ijerph-16-03054-t002:** Food Purchases, Preparation Practices, and Consumption.

	Baseline *n* = 1110	12-Month Follow-Up *n* = 1098	χ^2^ (*df*) or t
**Food Purchases**			
**Soy Sauce**			
Size of regular soy sauce purchased			0.82 (2)
Small bottle	258 (23.50%)	250 (23.02%)	
Regular bottle	693 (63.11%)	676 (62.25%)	
Gallon canister	147 (13.39%)	160 (14.37%)	
Regular soy sauce purchasing interval, month ^†^	1.86 (1.54), 0.25–6	2.02 (1.65), 0.25–6	t = −2.36 *
M(SD), range			
**Grain Products**			
White rice bag size			6.35 (2)
5 pounds or less	176 (16.62%)	212 (20.17%)	
10–35 pounds	626 (59.11%)	569 (54.14%)	
40 pounds or more	257 (24.27%)	270 (25.69%)	
White rice purchasing interval, month ^†^	1.94 (1.45), 0.25–6	2.11 (1.40), 0.5–6	t = −2.64 **
M(SD), range			
Brown rice bag size			51.94 (3) ***
Does not eat brown rice	265 (26.74%)	170 (17.97%)	
5 pounds or less	215 (31.79%)	349 (36.89%)	
10–35 pounds	376 (37.94%)	397 (41.97%)	
40 pounds or more	35 (3.53%)	30 (3.17%)	
Brown rice purchase interval, month ^†^	1.88 (1.38), 0.25–6	1.80 (1.24), 0.25–5	t = −1.19
M(SD), range			
**Oil**			
Types of oil purchased			
Canola	307 (27.66%)	400 (36.56%)	19.93 (1) ***
Sesame	155 (13.96%)	209 (19.10%)	10.55 (1) ***
Corn	238 (21.44%)	220 (20.11%)	0.59 (1)
Olive	432 (38.92%)	408 (37.29%)	0.62 (1)
Vegetable	202 (18.20%)	195 (17.82%)	0.05 (1)
Container size of oil purchased			3.97 (2)
Small bottle	155 (14.31%)	182 (17.28%)	
Regular bottle	688 (63.53%)	634 (60.21%)	
Gallon canister	240 (22.16%)	237 (22.51%)	
Oil purchase interval, month ^†^	1.91 (1.46), 0.25–6	2.31 (1.64), 0.5–6	t = −6.05 ***
M(SD), range			
**Reads nutrition label at purchase**			
Sugar	506 (53.66%)	563 (53.67%)	0 (1)
Sodium	406 (43.15%)	485 (46.15%)	1.81 (1)
Calories	354 (37.64%)	446 (42.48%)	5.2 (1) *
Fiber	173 (18.35%)	301 (28.67%)	29.20 (1) ***
Unsaturated fat	198 (21.00%)	232 (22.10%)	0.35 (1)
Trans Fat	245 (25.98%)	319 (30.38%)	4.74 (1) *
Saturated Fat	172 (18.28%)	194 (18.46%)	0.01 (1)
**Food Preparation**	7		
**Number of people to cook for**	3.17 (1.41), 1–6	3.11 (1.34), 1–6	t = 0.38
**Salt use**			
Does not measure salt use	830 (80.98%)	818 (80.43%)	0.10 (1)
Does measure salt use	195 (19.02%)	199 (19.57%)	
**Oil use**			
Does not measure oil use	777 (76.25%)	818 (80.27%)	4.85 (1) *
Does measure oil use	242 (23.75%)	201 (19.73%)	
**Trying to use less sodium in cooking**	848 (80.61%)	849 (80.40%)	0.01 (1)
**Trying to use less oil in cooking**	901 (85.48%)	896 (84.77%)	0.21 (1)
**Trying to use more whole grain in cooking**	692 (66.35%)	773 (73.13%)	11.46 (1) ***
**Consumption**			
**How often add salt to food at table**			16.90 (2) ***
Never	338 (30.98%)	396 (36.97%)	
Sometimes	600 (55.00%)	577 (53.87%)	
Always	153 (14.02%)	98 (9.15%)	

* *p* < 0.05, ** *p* < 0.01, *** *p* < 0.001, ^†^ winsorized at 95th percentile for baseline and 12-month follow-up separately.

**Table 3 ijerph-16-03054-t003:** Knowledge of Nutrition and Health Effects Related to Sodium, Oil, and Fiber.

	Baseline *n* = 1110	12-Month Follow-Up *n* = 1098	χ^2^ (*df*), or t
**Physical activity**			
How much should people with diabetes exercise			9.48 (1) **
Right (most days of the wk for 30+ min)	782 (72.34%)	840 (78.07%)	
Wrong (other answers)	299 (27.66%)	236 (21.93%)	
Health benefit of physical activity score,	0.64 (0.27), 0–1	0.63 (0.25), 0–1	t = 0.90
M(SD), range			
**Sodium**			
Recommended daily sodium intake			186.00 (1) ***
Right (one teaspoon)	129 (12.07%)	402 (37.54%)	
Wrong (other answers)	940 (87.93%)	669 (62.46%)	
Main source of sodium in American diet			5.10 (1) *
Right (processed foods)	374 (34.73%)	425 (39.42%)	
Wrong (other answers)	703 (65.27%)	653 (60.58%)	
**Oil**			
Healthiest type of fat			12.01 (1) ***
Right (monounsaturated and polyunsaturated)	448 (40.36%)	524 (47.68%)	
Wrong (other answers)	662 (59.64%)	575 (52.32%)	
Healthy types of oils score	0.47 (0.21), 0–1	0.50 (0.20), 0–1	t = −3.45
M(SD), range			
**Fiber**			
Carbohydrates with highest fiber score,	0.69 (0.27), 0–1	0.70 (.27), 0–1	t = 0.87
M(SD), range			
Health benefits of high fiber diet score,	0.49 (0.14), 0–0.86	0.49 (0.14), 0–0.86	t = 0
M(SD), range			
Knowledge of sodium, oil, and fiber score,	4.32 (1.22), 1.06–7.21	4.68 (1.35), 1.06–7.71	t = 1.29 ***
M (SD), range (0–8)			

* *p* < 0.05, ** *p* < 0.01, *** *p* <0.001.

**Table 4 ijerph-16-03054-t004:** Exposures to Health Promotions.

Participants’ Exposure to Health Promotions	Baseline *n* = 1110	12-Month Follow-Up *n* = 1098	χ^2^ (*df*)
Noticed signs that promoted healthier foods in general	372 (35.43%)	367 (35.32%)	0.002 (1)
Noticed signs that promoted healthier oil	296 (29.16%)	352 (44.39%)	44.88 (1) ***
Noticed signs that promoted whole grain	296 (31.56%)	342 (43.96%)	28.01 (1) ***
Noticed signs that promoted low-sodium products	266 (25.75%)	350 (34.09%)	17.05 (1) ***

*** *p* <0.001.

**Table 5 ijerph-16-03054-t005:** Outcome Measures: Healthier Purchasing Behaviors at 12-Month Follow-Up Survey.

Healthier Purchasing Behaviors	*n* (%)
Purchased low-sodium products in the past 6 months	419 (40.37%)
Used in cooking	299 (71.36%)
Family and respondent liked it	323 (77.09%)
Would continue to use it in the future	338 (80.67%)
Purchased whole grain products in the past 6 months	377 (37.48%)
Used in cooking	251 (68.71%)
Family and respondent liked it	280 (78.21%)
Would continue to use it in the future	302 (82.51%)
Purchased healthier oil in the past 6 months	378 (37.80%)
Used in cooking	290 (78.59%)
Family and respondent liked it	301 (82.02%)
Would continue to use it in the future	311 (83.38%)

**Table 6 ijerph-16-03054-t006:** Results of Multivariate Logistical Regression of Three Healthier Food Purchasing Behaviors at 12-Month Follow-Up Survey.

Regression Results	Purchased Low-Sodium Products (*n* = 851)	Purchased Whole Grain Products (*n* = 771)	Purchased Healthier Oil Products (*n* = 781)
	β (s.e.)	β (s.e.)	β (s.e.)
Knowledge score	0.24 (0.07) ***	0.33 (0.07) ***	0.35 (0.07) ***
Noticed signs that promoted healthier foods in general (ref: no)	0.59 (0.23) **	0.53 (0.32)	−0.55 (0.36)
Noticed signs that promoted less sodium	0.20 (0.24)	-	-
Noticed signs that promoted more whole grain	-	0.47 (0.33)	
Noticed signs that promoted healthier oil	-	-	1.58 (0.36) ***
Ethnicity (ref: Chinese)	1	1	1
Korean	0.19 (0.25)	−0.03 (0.26)	−0.005 (0.27)
Vietnamese	0.05 (0.23)	−0.54 (0.26) *	0.34 (0.25)
Filipino	2.14 (0.33) ***	2.15 (0.37) ***	1.93 (.35) ***
Other Asian	1.74 (0.77) *	1.04 (0.83)	1.28 (0.87)
Female (ref: male)	0.49 (0.17) **	0.57 (0.19) **	0.62 (0.19) **
Educational attainment (ref: < hs)	1	1	1
high school	−0.09 (0.24)	−0.31 (0.27)	−0.23 (0.26)
college or higher	0.33 (0.25)	−0.09 (0.28)	0.20 (0.28)
English proficiency (ref: poor/not well)	1	1	1
Well/fluent	−0.28 (0.20)	0.14 (0.22)	−0.03 (0.22)
Constant	−2.68	−3.02	−3.34
Log-likelihood chi-square (*df*)	157.53 (11) ***	193.12 (11) ***	187.13 (11) ***
McFadden R-square	0.13	0.19	0.18

Abbreviation: s.e. = standard error; * *p* < 0.05, ** *p* < 0.01, *** *p* < 0.001.

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
