# Peer review of "The Evaluation of IDEAL-REACH Program to Improve Nutrition among Asian American Community Members in the Philadelphia Metropolitan Area"

_ijerph, 2019, doi:10.3390/ijerph16173054_

Round 1

Reviewer 1 Report

Line 63-64—Reword and remove the repeated “from”.

Line 90-91—The description of sodium reductions in the review of literature are more than modest restrictions. Suggest removing “modest” from line 90.

Line 97-99. This is a awkward sentence. Please review.

Line 108—please elaborate how “individual food preparation” is part of a community-level interventions.

Results:

Table 2—Please add the significance annotations to table 2. You mention the P values in the narrative, but I did not see the significant differences indicated in the table.

Line 296-297.  How can you prove that the IDEAL campaign was the only campaign during this time?  Did you assess for other environmental campaigns?

Author Response

We sincerely appreciate the careful reading, editorial, and substantive reviews from the reviewers on our manuscript titled “The Evaluation of IDEAL Program to Improve Nutrition among Asian American CBO members in the Philadelphia Metropolitan Area.” We have provided below the point-by-point responses to the comments.

Please let us know if you have any further feedback. We look forward to hearing back from you.

Reviewer   1 Comments

Responses

Line 63-64—Reword and   remove the repeated “from”.

revised

Line 90-91—The   description of sodium reductions in the review of literature are more than   modest restrictions. Suggest removing “modest” from line 90.

revised

Line 97-99. This is an   awkward sentence. Please review.

revised

Line 108—please   elaborate how “individual food preparation” is part of the community-level   intervention.

We revised this section.

Results:

Table 2—Please add the   significance annotations to table 2. You mention the P values in the   narrative, but I did not see the significant differences indicated in the   table.

We added the p-values and annotations back in the table.

Line 296-297.    How can you prove that the IDEAL campaign was the only campaign during this   time?  Did you assess for other environmental campaigns?

We removed the contexts of interest here, and addressed the issue in   the discussion part later in the manuscript.

Reviewer 2 Report

This is an interesting paper that deals with an important topic namely community based dietary interventions.  However the paper suffer from some weaknesses that needs to be addressed

1.      The literature on community food interventions is not properly described. For instance Multi Level Multi component interventions MLMC, CBPR, Whole of Community interventions etc and specific program such as BHealthy Baltimore, Shape Up Somerville etc. Its important to understand how the current paper and is research fits into this research field where the gaps are and what the research adds to the existing body of knowledge. In that light I would consider rephrasing the conclusion in line 33 in abstract.

2.      It is not clear what understanding of community that is applied in the study? Is it something spatial in the sense that they are physical blocks or neigborhoods limited by geographical boundaries?  Or is community also something social/cultural in the sense that a CBO is considered a community? And that the members of that community can live different places in the city? In that way community would be less a geographical thing. Please explain.

3.      The idea of CBO’s plays an important role. The authors argue that these can function as kind of gatekeepers, intermediaries, mediators etc to facilitate and deliver the intervention components. But more detail is needed. How would the CBO approach differ from using other approaches. And how does the CBO approach differ from for instance the widely used WHO promoted settings approach? What assumptions are made on the gatekeepers/CBO leaders role in delivering the intervention components. For instance I would speculate that different CBO leaders would have different motivational levels for participating in and delivering the intervention components. I might be wrong but from reading the manuscript I get the impression that mediators, CBO leaders, chefs etc more or less “are doing as they are told”. More details on types and roles of these gatekeepers and the mechanisms that the authors assume will hold for these relationship – the researcher, CBO mediator , end user.

Minor

1.      Why only Asian Americans? Can one strategy do good for more types of residents? Or is the way that intervening through CBO’s you ONLY reach Asian Americans

2.      Line 83 “nuts, can lower blood pressure by 6/3 mm Hg. 23.”. Something seems to be wrong with the reference and or/format

3.      In line 136 the intervention components are listed. These seems to be quite general in terms of healthy eating. This is in sharp contrast with the whole introduction that focuses only on salt reduction. This needs to be aligned

4.      Line 137. The intervention components are listed but tin general terms. More detail is needed and measures of dose delivery, program reach etc is needed. How many? How often. What frequency? What participation? Just some of the questions that could be addressed. Consider putting all this in a table with 4 rows – one for each component

5.      Line 150 “The selected organization clusters were stratified based 150 on the four race/ethnicity or language groups” Please explain. What is a cluster? One CBO or all CBO’s within one ethnicity? Or something else? Please explain the stratification steps and the convenience steps and why these strategies were chosen

6.      Line 122 and other places. What does it mean to be a CBO member. Enrolled formally in the CBO – which I assume is some kind of an NGO. Or is it something informal. I speculate that would have an influence on how you recruit participants? Please explain.

7.      Line 155 “A 46-item questionnaire baseline survey and a 48-item 12-month follow-up survey questionnaire 155 were developed”. More detail is needed. Did the author develop that from scratch and if so how was it developed. I would speculate that the literature would be rich in existing already validated questionnaires would be there?

8.      Line 166 “Food Purchases and Preparation Practices. Because one important component of our intervention was promoting low-sodium soy sauce to replace regular soy sauce” Here again there is a mismatch between the focus on salt and the general healthy eating focus

9.      Line 181”assess participants’ knowledge” Which participants. Previously more types of participants have been presented including “end users” and chefs. Please clarify

10.   Line 467. The conclusion seems to be more a summary of results and extended discussion. I would suggest the conclusion to be brief and more to the point

11.   I would suggest that the authors write more on the policy implications of the study. In other wordls “Who should do what based on your findings”?

Author Response

We sincerely appreciate the careful reading, editorial, and substantive reviews from the reviewers on our manuscript titled “The Evaluation of IDEAL Program to Improve Nutrition among Asian American CBO members in the Philadelphia Metropolitan Area.” We have provided below the point-by-point responses to the comments.

Please let us know if you have any further feedback. We look forward to hearing back from you.

Reviewer 2 Comments

Responses

1.      The literature on community food interventions   is not properly described. For instance Multi Level Multi component   interventions MLMC, CBPR, Whole of Community interventions etc and specific   program such as BHealthy Baltimore, Shape Up Somerville etc. Its important to   understand how the current paper and is research fits into this research   field where the gaps are and what the research adds to the existing body of   knowledge. In that light I would consider rephrasing the conclusion in line   33 in abstract.

We have added references to previous community-level interventions on   nutrition and health behaviors.

We consider the sentence Line 33 to be an appropriate conclusion of   the findings of this study.

2.      It is not clear what understanding of   community that is applied in the study? Is it something spatial in the sense   that they are physical blocks or neigborhoods limited by geographical   boundaries?  Or is community also something social/cultural in the sense   that a CBO is considered a community? And that the members of that community   can live different places in the city? In that way community would be less a   geographical thing. Please explain.

The community in this study is referencing to the Asian American   neighborhoods and areas in the greater Philadelphia metropolitan areas, a   combination of sociocultural and geographic senses.

3.      The idea of CBO’s plays an important role. The   authors argue that these can function as kind of gatekeepers, intermediaries,   mediators etc to facilitate and deliver the intervention components. But more   detail is needed. How would the CBO approach differ from using other   approaches. And how does the CBO approach differ from for instance the widely   used WHO promoted settings approach? What assumptions are made on the   gatekeepers/CBO leaders role in delivering the intervention components. For   instance I would speculate that different CBO leaders would have different   motivational levels for participating in and delivering the intervention   components. I might be wrong but from reading the manuscript I get the   impression that mediators, CBO leaders, chefs etc more or less “are doing as   they are told”. More details on types and roles of these gatekeepers and the   mechanisms that the authors assume will hold for these relationship – the   researcher, CBO mediator , end user.

We added more details on the active roles of CBO leaders, chefs, staff,   and other community stakeholders.

Minor

1.      Why only Asian Americans? Can one strategy do   good for more types of residents? Or is the way that intervening through   CBO’s you ONLY reach Asian Americans

As we described in the introduction part, the Asian Americans have a   high burden of cardiovascular diseases, and the Asian American diet is high   in sodium and fat. Therefore, there is a urgent need to improve nutrition and   healthy eating in this community.

2.      Line 83 “nuts, can lower blood pressure by 6/3   mm Hg. 23.”. Something seems to be wrong with the reference and or/format

revised

3.      In line 136 the intervention components are   listed. These seems to be quite general in terms of healthy eating. This is   in sharp contrast with the whole introduction that focuses only on salt   reduction. This needs to be aligned

We revised this part.

4.      Line 137. The intervention components are   listed but tin general terms. More detail is needed and measures of dose   delivery, program reach etc is needed. How many? How often. What frequency?   What participation? Just some of the questions that could be addressed.   Consider putting all this in a table with 4 rows – one for each component

We added frequencies of the intervention components.

5.      Line 150 “The selected organization clusters   were stratified based 150 on the four race/ethnicity or language groups”   Please explain. What is a cluster? One CBO or all CBO’s within one ethnicity?   Or something else? Please explain the stratification steps and the   convenience steps and why these strategies were chosen

We clarified the sampling.

6.      Line 122 and other places. What does it mean   to be a CBO member. Enrolled formally in the CBO – which I assume is some   kind of an NGO. Or is it something informal. I speculate that would have an   influence on how you recruit participants? Please explain.

There were both formal and informal members of the CBOs, including   churches, temples, senior centers, and other kinds of community organizations.

7.      Line 155 “A 46-item questionnaire baseline   survey and a 48-item 12-month follow-up survey questionnaire 155 were   developed”. More detail is needed. Did the author develop that from scratch   and if so how was it developed. I would speculate that the literature would   be rich in existing already validated questionnaires would be there?

We did try to mimic some of the questions from the BRFSS but the   brevity of our surveys made that limited to basics like whether or not they   add salt to their meal.

8.      Line 166 “Food Purchases and Preparation   Practices. Because one important component of our intervention was promoting   low-sodium soy sauce to replace regular soy sauce” Here again there is a   mismatch between the focus on salt and the general healthy eating focus

The literature review has separate paragraphs for salt, oil and fiber   so we feel it is balanced. However, part of the reorganizing of the   discussion we kept a better balance between salt, oil and fiber. We found   much more research on salt in the literature than on fiber and studies of oil   are very limited (mostly to one diet). We believe our study contributes to   the field and that other studies should use our detailed look at the   complexity of “nutrition and diet” in a few questions. 

9.      Line 181”assess participants’ knowledge” Which   participants. Previously more types of participants have been presented   including “end users” and chefs. Please clarify

They are CBO members. We clarified that in the text.

10.   Line 467. The conclusion seems to be more a   summary of results and extended discussion. I would suggest the conclusion to   be brief and more to the point

We revised the conclusion part to be more precise and concise.

11.   I would suggest that the authors write more on   the policy implications of the study. In other wordls “Who should do what   based on your findings”?

We added policy implication in the discussion session.

Reviewer 3 Report

I hope that the authors will consider writing an article detailing the development, impact, implementation, and evaluation of the CBPR process with these ethnic/cultural populations. 

Author Response

We sincerely appreciate the careful reading, editorial, and substantive reviews from the reviewers on our manuscript titled “The Evaluation of IDEAL Program to Improve Nutrition among Asian American CBO members in the Philadelphia Metropolitan Area.” We have provided below the point-by-point responses to the comments.

Please let us know if you have any further feedback. We look forward to hearing back from you.

Reviewer 3 Comments

Responses

I hope that the   authors will consider writing an article detailing the development, impact,   implementation, and evaluation of the CBPR process with these ethnic/cultural   populations. 

We definitely have this manuscript in plan.

Reviewer 4 Report

It was a great approach to reveal the effect of community based intervention.

Although its strength, there are some weak point to overcome.

First, line 142-143, you used the convenient sampling to recruit participant, Can you show the responding rate at baseline and at F/U?

Second,, can you place section 2.5 before section 2.4.?

Third, can you reduce section 2.4? It is confused. You need to move some contents to section 2.2 (ex. line 217-226)

Fourth. 

1) the description of resuls was too complex. I think it contains the fact (ex, line 253-255) and the interpretation (ex, line 255-256). So you need to separate them. You need to describe the fact brefly in results section.

2) You have to add the p-value in table 2 and table 3.

3) I can not find the knowledge score in table 3. (line 284-286)

Fifth, 

1) the description of discussion is not structured. Some part of results and some part of conclusion are needed to move into discussion.

2) Why did you use the term 'low-income' in line 334?

In introdution and methods, you did not mention it.

Sixth, what is the conclusion of this study? I think current conclusion section contains discussion part and conclusion part. So you have to describe your conclusion in conclusion section.

Additionally, you have to correct some mistakes

Line 191. 7->8?  (You used eight questions)

Line 229. Student's t-test -. student's t-test

Line 260. white rice -> brown rice

Line 330. Table 6 -> Table 4 and you need to change table 6 to table 4 in text.

Author Response

We sincerely appreciate the careful reading, editorial, and substantive reviews from the reviewers on our manuscript titled “The Evaluation of IDEAL Program to Improve Nutrition among Asian American CBO members in the Philadelphia Metropolitan Area.” We have provided below the point-by-point responses to the comments.

Please let us know if you have any further feedback. We look forward to hearing back from you.

Reviewer 4 Comments

Responses

First, line 142-143,   you used the convenient sampling to recruit participant, Can you show the   responding rate at baseline and at F/U?

The responding rates   data were not available.

Second,, can you place   section 2.5 before section 2.4.?

revised

Third, can you reduce   section 2.4? It is confused. You need to move some contents to section 2.2   (ex. line 217-226)

Revised according to   suggestions.

Fourth. 

1) the description of   resuls was too complex. I think it contains the fact (ex, line 253-255) and   the interpretation (ex, line 255-256). So you need to separate them. You need   to describe the fact brefly in results section.

We revised it to make   the results more concise. We understand that our results are complex but we   are also committed to publishing as much of our data as possible because this   study is the first study to use most of our methods. Despite one of the   reviewers thinking that there must be a lot of information like we collected,   in fact such in-depth data has yet to be collected for any population.

2) You have to add the   p-value in table 2 and table 3.

Added.

3) I can not find the   knowledge score in table 3. (line 284-286)

They are on last role   of Table 3.

Fifth, 

1) the description of   discussion is not structured. Some part of results and some part of   conclusion are needed to move into discussion.

We revised the   discussion, added subheadings to streamline this section.

2) Why did you use the   term 'low-income' in line 334?

In introdution and   methods, you did not mention it.

“Low-Income” were   referenced in Line 46 and 53.

Sixth, what is the   conclusion of this study? I think current conclusion section contains   discussion part and conclusion part. So you have to describe your conclusion   in conclusion section.

We revised the   discussion, added subheadings to streamline this section.

Additionally, you have   to correct some mistakes

Line 191.   7->8?  (You used eight questions)

Line 229. Student's   t-test -. student's t-test

Line 260. white rice   -> brown rice

Line 330. Table 6   -> Table 4 and you need to change table 6 to table 4 in text.

No participants scored   a perfect 8, hence the highest knowledge score being 7.

“Student’s t-test” is   the correct spelling.

White rice was   correct, because our findings were showing that white rice was getting less   popular.

Table 6 was referenced   correctly.

Round 2

Reviewer 4 Report

 Most points are corrected or revised.

But two points are not changed.

1. Why did you use the   term 'low-income' in line 334?

In introdution and   methods, you did not mention it.

-> You answered 'Low-Income” were  referenced in Line 46 and 53'.

I can not find 'low-income' in line 46-53

You think Asian-American is all 'low-income'?

To avoid that, I want you to delete 'low-income' in line 334.

2. In line 265-266, you described 'large bags of white rice (40+ pounds)
266 became less popular (3.53% to 3.17%)'.

But in Table 3, this is for brown rice, NOT whit rice.

PLEASE, compare your description with table 3.

Author Response

We removed "low income" from the paper, and corrected the typo in Line 265.